# Probing the Limits of Embodied Spatial Planning in LLMs

## Abstract

Can the symbolic reasoning of Large Language Models (LLMs) extend to the physical world, or do they lack a fundamental "mind's eye" for grounded physical reasoning? This paper investigates this question by probing the ability of LLMs to reason about a dynamic physically-grounded environment. We introduce a novel methodology centered on indoor bouldering, a task that demands spatial imagination to (1) construct a mental environment from coordinates, (2) simulate an embodied agent's movement within that environment, and (3) adhere to physical constraints from the agent. Using our purpose-built dataset, EmbodiedPlan, which incorporates multiple agent profiles to test embodied reasoning, we challenge state-of-the-art LLMs (e.g., GPT-4o, Gemini Pro) to generate plans for different embodied agents. Our experiments reveal a consistent gap between syntactic fluency and physical plausibility: models can generate plans that are syntactically correct yet physically naive and poorly adapted to the agent's body. The results suggest that current LLMs possess a "brittle" mind's eye, capable of manipulating spatial symbols but lacking the grounded imagination required for true physical reasoning.

## 1 Introduction

A key frontier for artificial intelligence is moving beyond abstract, symbolic manipulation and toward physical grounding – the ability to connect reasoning to real-world physics, geometry, and spatial constraints. While Large Language Models (LLMs) have shown remarkable performance on reasoning and planning tasks (Wei et al., 2022; Kojima et al., 2024; Wei et al., 2024; Huang et al., 2023; Bismay et al., 2025), where most existing benchmarks focus on abstract puzzles (Valmeekam et al., 2023; Ding et al., 2024b; Chia et al., 2024), text-based games, or simulated environments (Puig et al., 2018; Huang et al., 2022), their proficiency often stems from mastering syntactic and semantic patterns in text, leaving their capacity for grounded physical reasoning an open question.

We argue that true physical intelligence requires a suite of fundamental cognitive capabilities that are not adequately measured by existing benchmarks. We identify and investigate three such abilities:

- *Spatial Imagination*: The ability to construct an internal mental model of a physical environment from symbolic descriptions and to dynamically simulate actions and their consequences.
- *Embodied Reasoning*: The ability to understand how an agent's physical characteristics (e.g., height) fundamentally reshape the problem space and constrain possible actions.
- *Constraint-Aware Compositional Planning*: The ability to generate sequences of compositional actions that accomplish a goal while respecting the physical limitations imposed by the agent and environment.

To probe these abilities, we introduce a methodology centered on indoor bouldering, which serves as a controlled environment for this challenge. Unlike a simple graph traversal problem like a maze, a bouldering route is a sparse set of points in a 2D space, requiring an agent to perform Constraint-Aware Path Creation by discovering a physically viable sequence of full-body movements. Success depends critically on all three abilities: imagining the body in space, respecting its limits, and planning trajectories under the constant constraint of gravity. To operationalize this probe, we introduce **EmbodiedPlan**, a dataset and evaluation framework built on the standardized MoonBoard system, each paired with annotated full-body symbolic action plans (Figure 1). A distinctive aspect

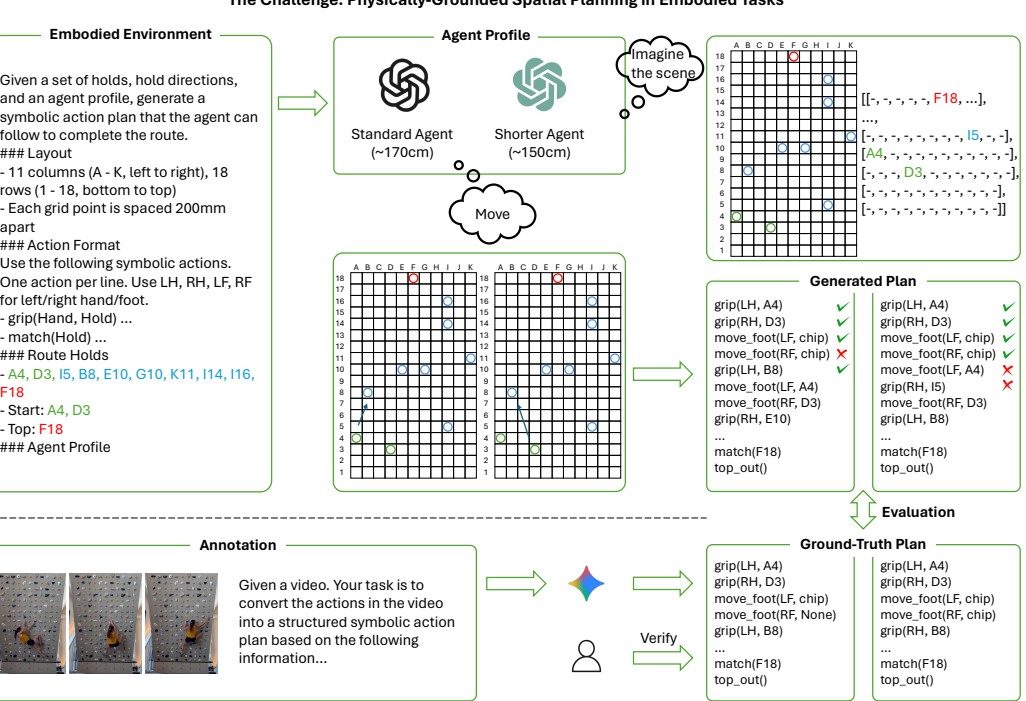

Figure 1: An overview of the EmbodiedPlan framework for probing the embodied spatial planning of LLMs. The process begins by providing the LLM with a bouldering problem (a set of specified holds: start holds, intermediate holds, and a final top hold) and an agent's physical profile (e.g., height). The LLM's task is to generate a plan as a sequence of symbolic actions to reach the final goal. The generated plan is then evaluated by comparing it against a human-in-the-loop annotated ground-truth plan to assess its symbolic correctness, semantic alignment, and its physical plausibility through center-of-gravity (CoG) trajectory simulation. Further implementation details are in the Appendix.

of EmbodiedPlan is the variation of agent embodiment to directly test for embodied reasoning. We model different agent profiles (e.g., short, medium, and tall climbers) and challenge models to adapt their plans to different physical abilities rather than generating a single, generic solution. For instance, a shorter agent may need an extra intermediate move that a tall agent can skip. This allows us to directly test whether an LLM can adapt its plan to an agent's unique physical capabilities and limitations. To assess performance, we design a comprehensive evaluation suite including symbolic correctness, semantic plan alignment, and a center-of-gravity (CoG) simulation to quantitatively assess the physical plausibility of the LLM's "imagined" trajectory.

Our experiments reveal that while LLMs can mimic the syntax of planning, their "mind's eye" is often blind to physical reality. The generated plans frequently contain spatially naive movements and demonstrate a poor grasp of embodied constraints, highlighting a critical deficit in their foundational abilities of spatial imagination and embodied reasoning. By diagnosing these failures, we aim to guide future research toward building LLMs that can reason about the world, not just over the text that describes it. In summary, our contributions are as follows:

- We introduce EmbodiedPlan, the first benchmark designed to evaluate dynamic, physically-constrained, and embodied planning in LLMs. It directly tests an LLM's ability to generate an actionable plan that respects geometric, physical, and bodily limitations.

- We incorporate physical variation of agent embodiment through agent profiles to test for adaptive, personalized planning.

- We design a validation framework including symbolic correctness, semantic plan alignment, and CoG trajectory simulation.

- We provide an extensive empirical study of state-of-the-art LLMs that highlights current limitations in physical reasoning and personalization, offering insights into how LLMs perform when grounded in embodied planning tasks.

## 2 RELATED WORK

The use of language models for planning in interactive and embodied environments has gained significant attention in recent years (Huang et al., 2022; Li et al., 2024; Du et al., 2024; Wu et al., 2023). Existing benchmarks provide structured evaluations of LLMs in planning tasks, but they often lack grounding in physical or embodied constraints. For instance, PlanBench (Valmeekam et al., 2023) focuses on reasoning about change via symbolic action sequences, while LoTa-Bench (Choi et al., 2024) benchmarks language-oriented task planners without fine-grained analysis of planning errors. The Embodied Agent Interface (Li et al., 2024) proposes a modular framework for evaluating LLMs across vision, action, and reasoning. While valuable, these benchmarks do not capture the complexities of domain-specific physical challenges. Recent work has also explored prompt engineering to mitigate hallucinations in path planning for LLMs (Deng et al., 2025), highlighting the challenges of grounding LLM outputs in spatial contexts.

A key challenge for LLMs is spatial reasoning. Several recent works have focused on benchmarking and improving this capability. "Mind the Gap" (Stogiannidis et al., 2025) and MANGO (Ding et al., 2024a) are benchmarks designed to evaluate spatial reasoning in vision-language models and the mapping and navigation abilities of LLMs, respectively. Other research has explored how to elicit spatial reasoning in LLMs through techniques like "Visualization-of-Thought" (Wu et al., 2024) and by studying how models can build spatial mental models from limited views Yin et al. (2025). These studies underscore the need for benchmarks that test spatial reasoning in complex, physically constrained scenarios.

In the domain of embodied AI, benchmarks in simulated domestic environments like ALFRED (Shridhar et al., 2020), which evaluates agents on everyday tasks, and VirtualHome (Puig et al., 2018), which models household activities via structured action programs, have been influential. However, they do not focus on the fine-grained physical constraints of a specialized domain like climbing. In the climbing domain, CIMI4D (Yan et al., 2023) introduces a multi-modal dataset aiming at 3D motion analysis. However, such datasets focus on physical movement reconstruction, rather than symbolic planning. EmbodiedPlan bridges this gap by introducing a benchmark for physically grounded, symbolic planning in the real-world domain of bouldering, offering a new dimension for assessing the embodied planning capabilities of LLMs and complementing existing benchmarks with a fresh challenge centered on physical plan feasibility.

## 3 A BOULDERING TASK TO PROBE FUNDAMENTAL ABILITIES

We designed a task environment and dataset, EmbodiedPlan, to serve as a rigorous testbed for the fundamental abilities of spatial imagination and embodied reasoning.

### 3.1 THE BOULDERING ENVIRONMENT: A 2D SPATIAL ENVIRONMENT

The environment for our task is the MoonBoard, a standardized training wall widely used in the climbing community, which is a 2D grid of bolt-on climbing holds arranged in 18 rows (numbered 1 to 18 from bottom to top) and 11 columns (labeled A to K) and set at a 40-degree overhanging angle. Each problem is defined by a subset of these holds: designated start holds (marked by green in the MoonBoard app), intermediate holds (marked blue), and a final top hold (marked red). For EmbodiedPlan, we curated a diverse set of problems from the official MoonBoard database in 2017 and 2019 settings, spanning difficulty grades from V3 to V9 in the V-grade, where higher numbers indicate greater complexity. Each hold's location is mapped to both a grid coordinate (e.g., "C10" refers to the hold at column C, row 10) and a 2D spatial coordinate, providing the symbolic and geometric information for the LLM's environment construction.

### 3.2 SYMBOLIC ACTION SPACE: A GRAMMAR OF MOVEMENT

To interface with LLMs, we developed a symbolic action space that functions as a compositional "grammar" of climbing movement. This vocabulary, informed by common climbing terminology, enables the model to deconstruct a continuous, full-body motion into a discrete, structured plan. The actions are: *grip(Hand, Hold)*: Move a specified hand (left or right) to a hold, e.g., *grip(LH, D4)*. *match(Hold)*: Move the other hand to the same hold, achieving two-hand control. *dynamic(Hand,*

*Hold)*: Execute a dynamic move (jump or lunge) to a distant hold. Feet are considered detached during this move. *move_foot(Foot, Hold)*: Place a foot (left or right) on a hold. This is the counterpart to *grip* for the lower limbs. The target can also be None to indicate lifting the foot off a hold. *top_out()*: Signal successful completion by controlling the top hold with both hands. We include the *top_out()* action at the end of every plan to explicitly mark the completion. Agents must begin with both hands on the start holds (or one hand on each, if two starts) and finish by controlling the top hold with both hands.

### 3.3 Agent Profiles: Testing Embodied Imagination

Another unique feature of EmbodiedPlan is its modeling of agent embodiment through the use of multiple agent profiles to test embodied reasoning. We collected ground-truth data from three climbers with distinct physical attributes, representing our agents:

- Agent 1 (Standard): A female climber of average height (∼170 cm).
- Agent 2 (Short): A shorter female climber (∼150 cm).
- Agent 3 (Tall): A taller male climber (∼180 cm).

These profiles are provided in the prompt to the LLM. As our analysis of human plans (§B.1) shows, these physical differences lead to measurably different climbing strategies. Plans differ across profiles to reflect physical feasibility – for example, a shorter agent may need to use an intermediate foothold to push up, where a taller agent can skip it. This setup challenges the LLM to condition its spatial imagination on the agent's embodiment and correctly infer its unique action affordance space.

### 3.4 Ground-Truth Plan Annotation

To construct a high-fidelity ground truth for our probe, we use a semi-automated, human-in-the-loop annotation pipeline designed for both efficiency and accuracy. This process begins by using a state-of-the-art vision-language model, Gemini 2.5 Pro, generating a first-pass annotation from YouTube videos processed at 5 frames per second (FPS). To ensure accuracy and consistency, we implement a two-stage verification process: (1) **Automatic Validation**: An automated script checks the plan for syntactic correctness and logical consistency. This included ensuring that all actions referenced valid holds within the problem set and that actions like *match* are used appropriately. (2) **Human Review**: The generated plans are further reviewed and corrected by our expert human annotators. The annotator's role is to refine the entire sequence to accurately match the technical movements and strategic nuances observed in the video. This pipeline, with comprehensive refinement and validation from automated quality checks and human experts, produces a robust ground-truth dataset of 400 problems annotated for the standard agent profile. From this collection, we create a specialized subset of 30 problems for which we have corresponding videos of three climbers with different physical characteristics. This subset is specifically used to evaluate personalized spatial planning and the models' capacity for adaptive embodied reasoning. Further implementation details, such as prompts and model specifications, are provided in the Appendix. Code and data are available here.

## 4 Evaluation

Our experiments probe the fundamental abilities of LLMs by tasking them with generating a symbolic plan for a specific route problem and agent profile. To diagnose the quality and limitations of their internal reasoning, our evaluation framework assesses generated plans across five dimensions: symbolic validity, plan-level characteristics, action overlap, sequence alignment, and spatial plausibility.

### 4.1 Evaluation Metrics

**(1) Validity (Syntactic & Semantic Correctness):** This metric serves as a baseline check for whether the LLM can adhere to the basic grammar of the task. We use a rule-based binary validator to check whether the generated plan adheres to the generation rules, symbolic action grammar, respects physical constraints, and is physically plausible. The validity check includes: **a. Format correctness (syntax check)**: All actions must conform to the predefined symbolic vocabulary and follow proper syntax (e.g., valid hold IDs, correct use of *match()*), with no unknown actions or free-form text. **b.**

**Route goal match (soft semantic check)**: The plan must begin with the designated start holds (start state) and end with a *top_out()* on the correct top hold (goal state). A plan is considered semantically correct if it starts on the correct start holds, uses only designated route holds, and ends with a *top_out()* on the goal hold – even if intermediate sequencing or foot placements differ. **c. Logical consistency**: The plan must respect physical common sense, such as only moving one limb at a time and avoiding unlikely sequences like more than two consecutive *grip()* actions without foot adjustments. We report the **validity rate** as the percentage of plans that satisfy all these constraints.

**(2) Plan-Level Characteristics:** To assess tendencies for under- or over-planning, we report the number of **actions** in each generated plan. We also compute the **normalized length**, defined as the number of actions divided by the number of holds in the problem, to account for route complexity.

**(3) Action Overlap (Compositional Accuracy):** To evaluate the correctness of the plan's content irrespective of strict ordering, we treat each plan as a bag of *(action_type, hold)* tokens and compute: **a. Precision:** the percentage of generated actions that match the ground truth. **b. Recall:** the percentage of ground-truth actions generated by the model. **c. F1 Score:** the harmonic mean of precision and recall, which is a reasonable proxy for "how close in content" the plans are. This metric emphasizes action and hold correctness over strict ordering and accommodates alternate but plausible plans that use the same critical holds.

**(4) Sequence Alignment:** Considering the sequence order of generated actions and measuring core overlap, we follow (Puig et al., 2018; Huang et al., 2022) and use: **a. Longest Common Subsequence (LCS):** the length of the longest ordered subsequence of actions shared between the generated and ground-truth plans. **b. Normalized LCS:** The LCS divided by the length of the ground-truth plan, allowing for fair comparison across problems of varying sequence lengths.

**(5) Spatial Plausibility (CoG Simulation):** To quantitatively evaluate the quality of the LLM's spatial imagination and assess physical plausibility of a generated plan, we simulate the trajectory of the agent's center-of-gravity (CoG) over the action sequence. We approximate the CoG at each step as the average of the coordinates of two hand positions. Specifically, for each problem, we store the spatial coordinates of all holds using a 2D coordinate system aligned with the standardized grid. Each hold is uniquely identified by its grid label (e.g., "G8" refers to column G, row 8) and mapped to Cartesian coordinates $(x, y)$, which are used to calculate distances between holds. For each plan, we track the CoG movement step by step and visualize the CoG trajectory: as the sequence progresses, we see how the CoG moves. We compute the total CoG displacement and compare the CoG trajectory of the generated plan to that of the ground-truth plan. Large deviations from the ground-truth trajectory or excessive movement suggest an inefficient, unstable, and physically naive plan, indicating a flawed internal simulation.

## 5 EXPERIMENTS AND ANALYSIS

To systematically diagnose the fundamental abilities of LLMs in a physically-grounded context, we conduct an extensive empirical study across a diverse suite of state-of-the-art models. This includes open-source families such as Llama, Qwen, Ministral, and Gemma, spanning from 3B to 70B parameters, as well as proprietary models like GPT, Gemini, Claude, and Grok. Our analysis, structured around our three central research questions, reveals that while models demonstrate basic syntactic fluency, they exhibit deficits in embodied reasoning and spatial imagination.

### 5.1 CAN LLMS DISTINGUISH SYNTACTIC CORRECTNESS FROM SPATIAL PLAUSIBILITY?

This question assesses whether LLMs are simply good at mimicking the format of a plan or if they understand its physical meaning. Our results, presented in Table 1, show a significant gap between a model's ability to follow syntactic rules and generate a spatially plausible and accurate plan.

Most modern LLMs, both open-source and proprietary, have become proficient at adhering to a specified grammar. Several models achieve high Validity scores, demonstrating strong syntactic competence. For example, Qwen3-4B (0.995) and Gemma3-12B (0.988) can almost flawlessly produce plans that conform to our action format and basic logical rules. The largest model, Llama-3.3-70B, also shows excellent instruction following with a validity of 0.965. This trend is solidified

Table 1: Performance of various LLMs on EmbodiedPlan for standard agent (Agent 1). We report on several key metrics: *Validity* (the proportion of syntactically and logically correct plans), *Actions* (the number of steps in the generated plan, compared to a human average of 17.0), and plan accuracy measured by *F1 Score* (action overlap) and *Normalized LCS* (sequence alignment).

| Model | Validity (↑) | Actions | Precision | Recall | F1 (↑) | LCS | Norm. LCS (↑) |
|---|---|---|---|---|---|---|---|
| Llama3.2-3B | 0.723 | 27.6 | 0.291 | 0.438 | 0.339 | 5.96 | 0.233 |
| Qwen2.5-3B | 0.555 | 10.9 | 0.352 | 0.215 | 0.260 | 3.39 | 0.199 |
| Qwen3-4B | 0.995 | 21.5 | 0.380 | 0.463 | 0.406 | 6.63 | 0.303 |
| Qwen2.5-7B | 0.895 | 14.9 | 0.338 | 0.287 | 0.304 | 4.42 | 0.250 |
| Llama3.1-8B | 0.830 | 37.7 | 0.281 | 0.495 | 0.344 | 7.36 | 0.248 |
| Ministral-8B | 0.860 | 16.9 | 0.292 | 0.275 | 0.276 | 4.40 | 0.236 |
| Gemma3-12B | 0.988 | 35.4 | 0.236 | 0.465 | 0.307 | 7.14 | 0.214 |
| Qwen3-30B | 0.168 | 21.2 | 0.373 | 0.438 | 0.396 | 6.79 | 0.329 |
| Llama3.3-70B | 0.965 | 19.8 | 0.403 | 0.462 | 0.427 | 6.78 | 0.339 |
| GPT-4o-mini | 0.940 | 18.6 | 0.362 | 0.375 | 0.360 | 5.75 | 0.298 |
| GPT-4.1-mini | 0.988 | 19.3 | **0.470** | **0.524** | **0.491** | 7.57 | **0.388** |
| GPT-5-mini | 1.000 | 18.2 | 0.437 | 0.458 | 0.443 | 7.12 | 0.378 |
| Gemini-2.5-flash | 0.505 | 20.1 | 0.433 | 0.501 | 0.460 | 7.68 | 0.382 |

by the latest proprietary models, with GPT-5-mini achieving a 1.000 validity score. This indicates that the challenge is not simply one of formatting the output correctly.

However, this syntactic proficiency does not translate to meaningful plan accuracy, which serves as our proxy for spatial plausibility. The plan accuracy scores, measured by F1 and Normalized LCS, are dramatically lower across the board. Among open-source models, Llama-3.3-70B achieves the highest F1 score (0.427) and normalized LCS (0.339). The proprietary models push this ceiling higher, with GPT-4.1-mini achieving an F1 score of 0.491 and a normalized LCS of 0.388. Despite this improvement, the fundamental gap persists: a plan that is 98.8% syntactically correct is still less than 50% accurate in its plan content and less than 40% aligned with a valid human sequence. This wide gap is the clearest evidence that the models can generate text that looks like a plan but lacks a deep understanding of the spatial and physical reasoning required to make the plan work.

**Analysis of Scaling Effects.** The results suggest a general, though imperfect, positive correlation between model size and planning capability. This is most evident within the Llama model family. As the model size increases from 3B to 8B to 70B, performance consistently improves across all key metrics: Validity increases from 0.723 to 0.965, the F1 score rises from 0.339 to 0.427, and the normalized LCS grows from 0.233 to 0.339. This strong trend indicates that spatial planning and reasoning are complex abilities that benefit significantly from increased model scale. The 70B model's better performance suggests it has developed a more sophisticated internal model.

**Precision vs. Recall and Planning Styles.** With a ground-truth average of 17.0 actions, the data reveals distinct and often flawed planning strategies:

- *A "Verbose" Strategy*: Models like Llama-3.1-8B (37.7 actions) and Gemma3-12B (35.4 actions) generate more than double the required number of steps. Their high recall (0.495 and 0.465, respectively) and very low precision (0.281 and 0.236) confirm they are employing an approach that produces an exhaustive list of moves in the hope of including the correct ones, which sacrifices the plan's coherence and efficiency.

- *A "Conservative" Strategy*: Qwen2.5-3B (10.9 actions) exemplifies under-planning, producing overly simplistic plans that miss critical moves, as reflected by its low recall of 0.215.

- *A "Balanced" Strategy*: The top-performing open-source model, Llama-3.3-70B and proprietary models demonstrate a more advanced approach. Their action counts (ranging from 18.2 to 20.1) are much closer to the human baseline. GPT-4.1-mini, for instance, has a well-balanced precision (0.470) and recall (0.524), leading to its top-performing F1 score (0.491).

Performance is not purely a function of size, and certain models exhibit unique behaviors. The Qwen family shows notable inconsistencies. The Qwen3-4B model is a standout performer for its size, achieving an F1 score (0.406) and normalized LCS (0.303) that are highly competitive. Conversely, the Qwen3-30B model presents a significant anomaly: despite achieving a strong normalized LCS (0.329), its validity score is catastrophically low at 0.168. This highlights that reasoning capabilities must be matched by reliable instruction-following.

Table 2: Personalized planning performance of LLMs on EmbodiedPlan across three agent profiles: Agent 1 (standard), Agent 2 (short), and Agent 3 (tall). Plan divergence is measured by *Norm. LCS → Agent 1*, where a lower score indicates stronger personalization, with the human baselines for Agent 2 (0.641) and Agent 3 (0.754) serving as a reference for effective adaptation. Colored subscripts on the *F1* and *Norm. LCS* scores indicate the change in plan accuracy relative to Agent 1 (green for improvement, red for decline).

| Agent | Model | Validity (↑) | Actions | Precision | Recall | F1 (↑) | LCS | Norm. LCS (↑) | LCS → Agent 1 | Norm. LCS → Agent 1 |
|---|---|---|---|---|---|---|---|---|---|---|
| | Human | 1.0 | 21.7 | – | – | – | – | – | – | – |
| | Llama-3.2-3B | 0.4 | 24.7 | 0.255 | 0.266 | 0.255 | 4.8 | 0.183 | – | – |
| | Qwen2.5-3B | 0.7 | 11.7 | 0.323 | 0.165 | 0.210 | 3.3 | 0.150 | – | – |
| | Qwen3-4B | 0.7 | 23.4 | 0.294 | 0.305 | 0.292 | 5.6 | 0.214 | – | – |
| | Qwen2.5-7B | 0.7 | 19.8 | 0.254 | 0.229 | 0.240 | 4.1 | 0.180 | – | – |
| | Llama-3.1-8B | 0.7 | 31.5 | 0.274 | 0.374 | 0.310 | 7.3 | 0.235 | – | – |
| Agent 1 | Ministral-8B | 0.7 | 22.7 | 0.300 | 0.286 | 0.289 | 5.9 | 0.241 | – | – |
| | Gemma3-12B | 0.7 | 32.1 | 0.367 | 0.510 | 0.417 | 9.1 | 0.297 | – | – |
| | Qwen3-30B | 0.2 | 18.8 | 0.424 | 0.350 | 0.381 | 7.1 | 0.313 | – | – |
| | Llama-3.3-70B | 0.7 | 18.5 | 0.426 | 0.358 | 0.388 | 6.4 | 0.288 | – | – |
| | GPT-4o | 0.8 | 18.7 | 0.628 | 0.531 | 0.574 | 9.5 | 0.438 | – | – |
| | Gemini | 0.6 | 23.0 | 0.611 | 0.636 | 0.619 | 9.7 | 0.407 | – | – |
| | Claude | 0.4 | 19.6 | 0.547 | 0.493 | 0.516 | 9.0 | 0.411 | – | – |
| | Grok | 0.8 | 20.6 | 0.652 | 0.613 | 0.628 | 9.2 | 0.408 | – | – |
| | Human | 1.0 | 24.9 | – | – | – | – | – | 16.0 | 0.641 |
| | Llama-3.2-3B | 0.6 | 26.4 | 0.259 | 0.260 | 0.254 $_{-0.001}$ | 5.0 | 0.175 $_{-0.008}$ | 18.9 | 0.737 |
| | Qwen2.5-3B | 0.7 | 12.8 | 0.338 | 0.171 | 0.223 $_{+0.013}$ | 3.8 | 0.153 $_{+0.003}$ | 11.1 | 0.894 |
| | Qwen3-4B | 0.7 | 24.9 | 0.291 | 0.292 | 0.282 $_{-0.009}$ | 5.7 | 0.190 $_{-0.024}$ | 22.7 | 0.930 |
| | Qwen2.5-7B | 0.7 | 19.4 | 0.258 | 0.210 | 0.229 $_{-0.011}$ | 4.6 | 0.183 $_{+0.003}$ | 17.7 | 0.886 |
| | Llama-3.1-8B | 0.7 | 31.7 | 0.317 | 0.386 | 0.340 $_{+0.029}$ | 8.6 | 0.263 $_{+0.028}$ | 27.4 | 0.874 |
| Agent 2 | Ministral-8B | 0.7 | 22.7 | 0.331 | 0.285 | 0.303 $_{+0.014}$ | 6.6 | 0.254 $_{+0.012}$ | 21.7 | 0.958 |
| | Gemma3-12B | 0.7 | 33.7 | 0.360 | 0.451 | 0.394 $_{-0.024}$ | 9.1 | 0.284 $_{-0.013}$ | 28.9 | 0.864 |
| | Qwen3-30B | 0.2 | 21.0 | 0.408 | 0.343 | 0.368 $_{-0.012}$ | 7.4 | 0.284 $_{-0.029}$ | 11.9 | 0.575 |
| | Llama-3.3-70B | 0.7 | 16.8 | 0.435 | 0.301 | 0.351 $_{-0.037}$ | 6.0 | 0.241 $_{-0.046}$ | 12.2 | 0.667 |
| | GPT-4o | 0.8 | 19.0 | 0.645 | 0.485 | 0.551 $_{-0.023}$ | 10.0 | 0.408 $_{-0.030}$ | 13.9 | 0.722 |
| | Gemini | 0.6 | 23.5 | 0.649 | 0.608 | 0.620 $_{+0.001}$ | 11.4 | 0.439 $_{+0.032}$ | 11.4 | 0.463 |
| | Claude | 0.4 | 20.8 | 0.581 | 0.486 | 0.527 $_{+0.011}$ | 9.8 | 0.396 $_{-0.015}$ | 15.0 | 0.706 |
| | Grok | 0.8 | 18.5 | 0.680 | 0.503 | 0.576 $_{-0.052}$ | 10.9 | 0.438 $_{+0.030}$ | 14.9 | 0.729 |
| | Human | 1.0 | 19.9 | – | – | – | – | – | 16.5 | 0.754 |
| | Llama-3.2-3B | 0.4 | 20.2 | 0.278 | 0.250 | 0.258 $_{+0.004}$ | 4.8 | 0.221 $_{+0.038}$ | 15.3 | 0.632 |
| | Qwen2.5-3B | 0.7 | 12.8 | 0.288 | 0.174 | 0.213 $_{+0.003}$ | 3.2 | 0.159 $_{+0.009}$ | 10.9 | 0.871 |
| | Qwen3-4B | 0.7 | 25.7 | 0.255 | 0.325 | 0.279 $_{-0.013}$ | 5.6 | 0.202 $_{-0.012}$ | 22.4 | 0.884 |
| | Qwen2.5-7B | 0.7 | 19.7 | 0.237 | 0.233 | 0.234 $_{-0.006}$ | 4.3 | 0.204 $_{+0.024}$ | 18.0 | 0.896 |
| | Llama-3.1-8B | 0.7 | 31.8 | 0.275 | 0.418 | 0.326 $_{+0.016}$ | 7.6 | 0.246 $_{+0.011}$ | 25.9 | 0.787 |
| Agent 3 | Ministral-8B | 0.7 | 23.3 | 0.267 | 0.291 | 0.276 $_{-0.013}$ | 5.7 | 0.242 $_{0.000}$ | 21.5 | 0.924 |
| | Gemma3-12B | 0.7 | 32.5 | 0.320 | 0.488 | 0.381 $_{-0.036}$ | 8.0 | 0.254 $_{-0.043}$ | 29.9 | 0.922 |
| | Qwen3-30B | 0.2 | 19.5 | 0.417 | 0.380 | 0.396 $_{+0.015}$ | 7.0 | 0.330 $_{+0.017}$ | 14.4 | 0.738 |
| | Llama-3.3-70B | 0.7 | 18.4 | 0.403 | 0.372 | 0.381 $_{-0.007}$ | 5.6 | 0.265 $_{-0.023}$ | 13.2 | 0.688 |
| | GPT-4o | 0.7 | 19.4 | 0.602 | 0.571 | 0.583 $_{+0.009}$ | 9.8 | 0.473 $_{+0.035}$ | 13.4 | 0.679 |
| | Gemini | 0.7 | 21.0 | 0.626 | 0.661 | 0.641 $_{+0.022}$ | 9.9 | 0.460 $_{+0.053}$ | 12.2 | 0.510 |
| | Claude | 0.4 | 20.2 | 0.498 | 0.500 | 0.497 $_{-0.019}$ | 8.8 | 0.416 $_{+0.005}$ | 14.0 | 0.686 |
| | Grok | 0.6 | 18.7 | 0.579 | 0.538 | 0.557 $_{-0.071}$ | 9.0 | 0.439 $_{+0.031}$ | 13.4 | 0.653 |

## 5.2 CAN LLMs ADAPT PLANS TO AN AGENT'S EMBODIMENT?

The capacity for embodied reasoning – adapting a plan to an agent's physical form – is a critical test of grounded intelligence. Our analysis reveals that this is a nuanced capability, largely absent in most open-source models but emerging at scale, with only the most advanced closed-source models demonstrating it robustly. To establish a benchmark for this task, we first analyzed the human ground truth, which confirms that physical embodiment dictates strategy. The shorter human agent requires a significantly different plan than the standard agent, with a plan divergence (Norm. LCS → agent 1) of 0.641.

**Embodied Reasoning is Largely Absent in Most Open-Source Models.** The majority of the open-source models tested fail the embodied reasoning task. As shown in Table 2, models like Ministral-8B and Qwen3-4B show almost no adaptation to the agent's profile. Their divergence scores (Norm. LCS → Agent 1) for the shorter Agent 2 are 0.958 and 0.930, respectively. This means the plans they generate are over 90% identical to their plans for the standard agent a, indicating they largely ignore the embodiment information in the prompt. In addition, their low plan accuracy scores remain stagnant across all profiles. This indicates not just a failure to personalize, but a general inability to form accurate plans for any agent.

**Adaptation Appears as an Emergent but Flawed Capability at Scale.** This critical reasoning ability appears to be an emergent property at scale, though its implementation in the largest open-

source models remains flawed. Qwen3-30B and Llama-3.3-70B show a remarkable ability to adapt their plan structure, with divergence scores for Agent 2 of 0.575 and 0.667, respectively, closely mirroring the human baseline of 0.641. However, this adaptation is not effective. When Llama3.3-70B personalizes its plan for the more difficult shorter agent, its accuracy significantly decreases, with the F1 score dropping from 0.388 to 0.351. Furthermore, it fails a simple physical heuristic, incorrectly generating fewer actions for the shorter agent (16.8) than for the taller agent (18.4). This suggests the model knows it must change its plan but lacks the grounded understanding to change it correctly, e.g., it fails to grasp the basic physical implication that a shorter agent often needs more intermediate moves, resulting in a different but objectively worse plan.

**Closed-Source Models Show More Robust, but Still Imperfect, Adaptation.** In contrast, state-of-the-art closed-source models consistently demonstrate a more robust capacity for embodied reasoning, although their performance reveals different levels of sophistication and their own set of imperfections. While all four models adapt their plan structures, models like GPT-4o and Grok do so ineffectively. While clearly personalizing their plans (divergence scores of 0.722 and 0.729 for Agent 2, respectively), they produce adapted plans that are less accurate than their standard ones. GPT-4o's F1 score drops from 0.574 for Agent 1 to 0.551 for Agent 2; Grok's drops from 0.628 to 0.576. Furthermore, they both fail the same plan length heuristic as Llama3.3-70B, incorrectly generating fewer actions for the shorter agent. Like the largest open-source models, they adapt, but the adaptation is not fully grounded in physical reality. Gemini, however, stands out as the only model that demonstrates true, effective embodied reasoning across all metrics. It exhibits the strongest adaptation signal (with a divergence score of 0.463 for Agent 2), correctly intuits the need for a longer plan for Agent 2 (generating more actions for Agent 2 (23.5) than Agent 1 (23.0) or Agent 3 (21.0)), and, most importantly, its adapted plans become more accurate. The F1 score increases from 0.619 for Agent 1 to 0.620 for Agent 2 and 0.641 for Agent 3. The normalized LCS shows a similar trend, improving from 0.407 to 0.439 for Agent 2. This indicates that Gemini can consistently translate a change in embodiment into a different and objectively better plan.

### 5.3 CAN THE FLAWS IN AN LLM'S SPATIAL IMAGINATION BE QUANTIFIED?

By treating the generated plan as an external projection of the LLM's internal simulation, we use center-of-gravity (CoG) analysis to quantitatively and qualitatively probe the flaws in its implicit physical world model, providing a window into the LLM's "mind's eye". Quantitatively, the CoG path lengths (Figure 2) reveal flawed reasoning: for standard and tall agents, most LLM-generated paths result in a greater total CoG displacement than the human ground-truth plans, indicating physically inefficient and redundant imagined movements. In contrast, for the shorter Agent 2, all models generate shorter CoG trajectories than the human reference, indicating that the models may generate plans that the agent finds hard to execute. Among the evaluated models, Gemini 2.5 Pro generates the CoG path length most closely aligned with the human trajectory for Agent 3, demonstrating a better ability to produce physically realistic plans. Overall, these results suggest that while current LLMs show signs of agent embodiment adaptation, there is still room for improvement in generating movement plans that align with natural, human-like body mechanics.

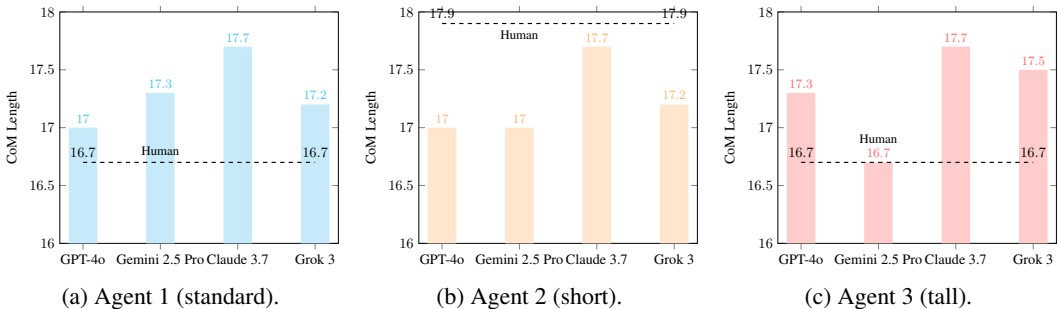

(a) Agent 1 (standard).      (b) Agent 2 (short).      (c) Agent 3 (tall).

Figure 2: Comparison of center-of-gravity (CoG) trajectory lengths between LLM-generated plans (bars) and the human ground-truth (dashed line) across three agent profiles. For Agents 1 and 3, most LLM plans are less efficient (longer path) than the human benchmark. Conversely, for the shorter Agent 2, all LLMs generate overly simplistic plans with shorter paths, suggesting a failure to account for necessary stabilizing movements.

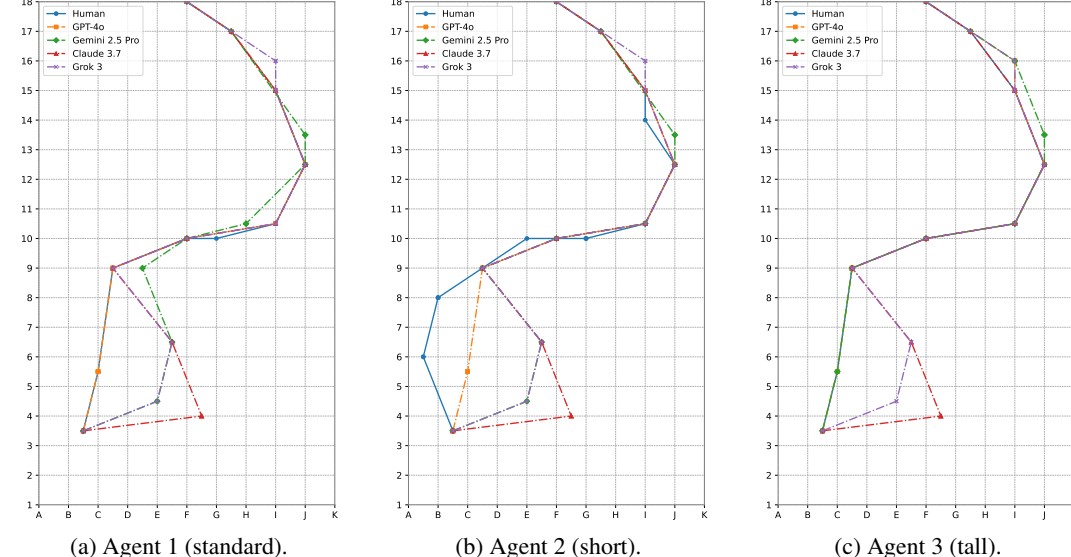

(a) Agent 1 (standard).      (b) Agent 2 (short).      (c) Agent 3 (tall).

Figure 3: Visualization of Center-of-Gravity (CoG) trajectories for a single complex problem, comparing the planning strategies of various LLMs against the human ground-truth (solid line). Each panel corresponds to a different agent profile.

Table 3: A case study of model performance on a single challenging route, comparing proprietary LLMs to the human ground truth across the three agent profiles.

| Model | Agent 1 | | | | Agent 2 | | | | Agent 3 | | | |
|---|---|---|---|---|---|---|---|---|---|---|---|---|
| | Actions | F1 (↑) | Norm. LCS (↑) | CoG Length | Actions | F1 (↑) | Norm. LCS (↑) | CoG Length | Actions | F1 (↑) | Norm. LCS (↑) | CoG Length |
| Human | 26 | – | – | 22.2 | 28 | – | – | 23.3 | 26 | – | – | 22.1 |
| GPT-4o | 25 | 0.667 | 0.500 | 22.1 | 25 | 0.604 | 0.464 | 22.1 | 26 | 0.615 | 0.500 | 22.4 |
| Gemini 2.5 Pro | 29 | 0.473 | 0.345 | 22.8 | 27 | 0.473 | 0.321 | 24.6 | 28 | 0.630 | 0.464 | 22.4 |
| Claude 3.7 Sonnet | 24 | 0.440 | 0.269 | 26.5 | 24 | 0.500 | 0.393 | 26.5 | 26 | 0.423 | 0.423 | 26.5 |
| Grok 3 | 27 | 0.604 | 0.259 | 24.8 | 27 | 0.655 | 0.571 | 24.8 | 24 | 0.600 | 0.538 | 24.8 |

**Case Study: Visualizing Route-Specific Planning.** To qualitatively illustrate the models' planning behaviors, we present a case study on a single, complex problem featuring an above-average number of holds: A4, D3, I5, B8, E10, G10, K11, I14, I16, F18. The problem begins with two hands split on A4 and D3 and ends with matched hands on F18 (visualized in Figure 3.) The human trajectories (solid lines) demonstrate effective embodied reasoning: the path for the shorter Agent 2 is visibly more gradual and longer, reflecting the necessary adaptations for their physical profile. While most CoG trajectories generated by LLMs are visibly divergent from the human baseline, providing visual proof of a poor mental simulation that fails to account for embodied reasoning, GPT-4o's plan for Agent 1 closely mirrors the human's trajectory. This is also supported by its high F1 score (0.667), normalized LCS (0.500), and a nearly identical CoG path length (Table 3), highlighting the performance gap between it and other models.

## 6 LIMITATIONS AND NEXT STEPS

In this work, we probe the fundamental limits of LLMs on physically grounded tasks using our EmbodiedPlan benchmark. Our findings reveal a critical gap between the models' syntactic fluency and the embodied spatial reasoning required for real-world interaction. The combined evidence suggests that simply scaling current architectures on more text data may be insufficient to achieve true physical intelligence. Research into architectures that can learn and maintain more explicit and robust world models is critical. Furthermore, training methodologies that better ground language in geometric and spatial principles could help bridge the gap we have identified. A more advanced paradigm would be to develop interactive refinement loops, where a plan generated by an LLM is executed in a simulator and the model uses success, failure, and feedback signals to iteratively correct its strategy, like reinforcement learning. By providing a challenging and quantifiable testbed, EmbodiedPlan can serve as a valuable tool for driving and measuring progress in these future explorations of embodied intelligence in AI.

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

# APPENDIX

## A EXPERIMENTAL SETUP

Our experimental framework, EmbodiedPlan, is designed to probe the fundamental abilities of LLMs in a controlled setting, as illustrated in Figure 1. We evaluate a diverse suite of state-of-the-art models on this benchmark.

### A.1 MODELS EVALUATED

We evaluate a wide range of LLMs to understand how these capabilities vary across different architectures and scales. This includes open-source families (Llama, Qwen, Ministral, and Gemma) spanning from 3B to 70B parameters, as well as proprietary models like GPT-4o, Gemini Pro, Claude, and Grok. A detailed list of all open-source models, their sources, and licenses is provided in Table 4.

### A.2 LLM PROMPT FOR EMBODIED REASONING

For each problem, the LLM is tasked with generating a complete, symbolic climbing plan based on a given route and a specific agent profile. To guide the models, we use a detailed prompt that encodes the route's spatial configuration and the agent's physical profile, explicitly conditioning the model to reason under embodiment constraints. The full prompt is shown in Box A.2. This prompt encodes the route's spatial configuration and agent profile, ensuring the model reasons under embodiment constraints. For all open-source LLMs, we set the temperature to 0 for reproducibility and max_new_token to 1024 to ensure complete outputs.

Table 4: Open-source Models.

| Model | Link | License |
|---|---|---|
| LLAMA-3.2-3B | https://huggingface.co/meta-llama/Llama-3.2-3B-Instruct | Llama 3.2 Community License |
| QWEN2.5-3B | https://huggingface.co/Qwen/Qwen2.5-3B-Instruct | qwen-research |
| QWEN3-4B | https://huggingface.co/Qwen/Qwen3-4B-Instruct-2507 | Apache license 2.0 |
| QWEN2.5-7B | https://huggingface.co/Qwen/Qwen2.5-7B-Instruct | Apache license 2.0 |
| LLAMA-3.1-8B | https://huggingface.co/meta-llama/Llama-3.1-8B-Instruct | Llama 3.1 Community License |
| MINISTRAL-8B | https://huggingface.co/mistralai/Ministral-8B-Instruct-2410 | mrl |
| GEMMA3-12B | https://huggingface.co/google/gemma-3-12b-it | Gemma |
| QWEN3-30B | https://huggingface.co/Qwen/Qwen3-30B-A3B-Instruct-2507 | Apache license 2.0 |
| LLAMA-3.3-70B | https://huggingface.co/meta-llama/Llama-3.3-70B-Instruct | Llama 3.3 Community License |

**Agent 1**

```
- Height: 172 cm
- Ape index: +0
- Gender: Female
```

**Agent 2**

```
- Height: 152 cm
- Ape index: +0
- Gender: Female
```

**Agent 3**

```
- Height: 179 cm
- Ape index: +0
- Gender: Male
```

Figure 4: Agent profiles which are corresponding to the climber phisical characteritics in the video.

### A.3 AGENT PROFILES

To directly test for embodied reasoning, our benchmark incorporates three distinct agent profiles with varying physical characteristics, which are based on the climbers in our source videos. These profiles, detailed in Figure 4, are defined by structured metadata including height, arm span (ape index), and gender. This metadata is explicitly included in the model's prompt, conditioning the LLM to generate a plan that respects the agent's body-specific limitations and unique action affordances. This experimental design allows us to evaluate whether an LLM can perform true personalization – for instance, by correctly generating extra intermediate moves for a shorter agent that a taller agent could skip.

### A.4 VLM PROMPT FOR DATA ANNOTATION

As part of our semi-automated data annotation pipeline, we utilize a Vision-Language Model (VLM), Gemini 2.5 Pro, to generate an initial draft of the symbolic plans. The model is prompted to produce a sequence of symbolic actions directly from the visual input of our source videos, which are processed at 5 frames per second (FPS). The complete prompt used for this task is provided in Box A.4, and a sample of the video annotation is shown in Figure 5. All source videos were obtained from YouTube and are licensed under Creative Commons CC BY.

## B MORE RESULTS

### B.1 ANALYSIS OF PERSONALIZED HUMAN PLANS

An analysis of the ground-truth data from the three human agents confirms that embodiment is not a minor detail but a primary driver of planning strategy. As shown in Table 5, agents with different physical profiles produce measurably different plans to solve the same problems.

The most significant factor is the agent's height, which directly impacts their reach and the number of actions required. The shorter agent (Agent 2) consistently takes more steps, with the highest average total actions (24.9) and normalized actions (3.5) per route. This aligns with the intuition that shorter climbers must perform additional, granular foot placements to reach the same handholds as their taller counterparts. In contrast, the taller agent (Agent 3) leverages greater reach to complete routes with the fewest actions on average (19.9).

These differences go beyond simple plan length and reflect fundamentally different strategies. By comparing the action sequences of the shorter and taller agents to the standard agent using the Normalized Longest Common Subsequence (LCS), we can quantify this strategic divergence. The shorter agent's plans show the most significant variation, with a normalized LCS of just 0.641 when

---

**Prompt for LLM planning**

You are a climbing expert. Given a set of MoonBoard climbing holds, hold directions, and a climber profile, generate a symbolic action plan that the agent can follow to complete the route.
### MoonBoard Layout
- 11 columns (A - K, left to right), 18 rows (1 - 18, bottom to top)
- Each grid point is spaced 200mm apart
### Rules
- The climb starts with both hands on the designated start hold(s). If only one start hold is provided, the agent starts with both hands matched on it.
- The climb ends on the designated finish hold(s). If there is only one finish hold, both hands must match on it.
- Feet may start on any kickboard chips.
- During the climb, feet follow hands and must only use marked holds or the board.
- The climb always ends with the action top_out().
### Action Format Use the following symbolic actions. One action per line. Use LH, RH, LF, RF for left/right hand/foot.
- grip(Hand, Hold): Move a hand (LH or RH) to a hold and grip it. Example: grip(LH, D4)
- match(Hold): Bring the other hand to the same hold currently held by one handhold. Example: match(D4)
- dynamic(Hand, Hold): Make a dynamic (jump/lunge) move to a far hold with one hand. Both feet are temporarily removed from the holds. Example: dynamic(RH, F15)
- move_foot(Foot, Hold): Move a foot (LF or RF) to a specific hold or kickboard chip or None to indicate free foot or smear. Example: move_foot(RF, F8), move_foot(LF, chip), move_foot(LF, None)
- top_out() – Mark the completion of the climb.
### Route Holds
- F4, I8, H12, I15, J18
- Start: F4
- Top: J18
### Hold Directions
- F4: N
- I8: N
- H12: N
- I15: N
- J18: W
### Climber Profile
- Height: 172 cm
- Ape index: +0
- Gender: Female
—
Using the provided holds, rules, and agent profile, generate a step-by-step symbolic beta plan.
- Begin with a valid dual-hand starting position
- End with top_out()
- Include one action per line
- Do not include any commentary or explanation

---

compared to the standard agent's plans. This indicates that nearly 36% of the actions are different, reflecting the major modifications needed to compensate for a more limited Action Affordance Space.

### B.2 ABLATION: HAND-ONLY PERFORMANCE

To better isolate the challenge of full-body coordination, we conducted an analysis on a simplified, hands-only version of the task, where all foot-placement actions are ignored (Tables 6, 7, and 8). Overall, LLMs perform significantly better on hand-only evaluations. This is expected, as hand actions are fewer, more visually salient, and follow clearer sequential patterns, making them easier

**Prompt for VLM planning**

You are a climbing expert. You are given a climbing video. Your task is to convert the climbing actions in the video into a structured symbolic action plan based on the following information:
### MoonBoard Layout
- 11 columns (A - K, left to right), 18 rows (1 - 18, bottom to top)
- Each grid point is spaced 200mm apart
### Rules
- The climb starts with both hands on the designated start hold(s). If only one start hold is provided, the agent starts with both hands matched on it.
- The climb ends on the designated finish hold(s). If there is only one finish hold, both hands must match on it.
- Feet may start on any kickboard chips.
- During the climb, feet follow hands and must only use marked holds or the board.
- The climb always ends with the action top_out().
### Action Format Use the following symbolic actions. One action per line. Use LH, RH, LF, RF for left/right hand/foot.
- grip(Hand, Hold): Move a hand (LH or RH) to a hold and grip it. Example: grip(LH, D4)
- match(Hold): Bring the other hand to the same hold currently held by one handhold. Example: match(D4)
- dynamic(Hand, Hold): Make a dynamic (jump/lunge) move to a far hold with one hand. Both feet are temporarily removed from the holds. Example: dynamic(RH, F15)
- move_foot(Foot, Hold): Move a foot (LF or RF) to a specific hold or kickboard chip or None to indicate free foot or smear. Example: move_foot(RF, F8), move_foot(LF, chip), move_foot(LF, None)
- top_out() – Mark the completion of the climb.
### Route Holds
- A2, B5, B8, E11, C14, F16, D18
- Start: A2, B5
- Top: D18
—
Using the provided holds, rules, and agent profile, generate a step-by-step symbolic beta plan.
- Begin with a valid dual-hand starting position
- End with top_out()
- Include one action per line

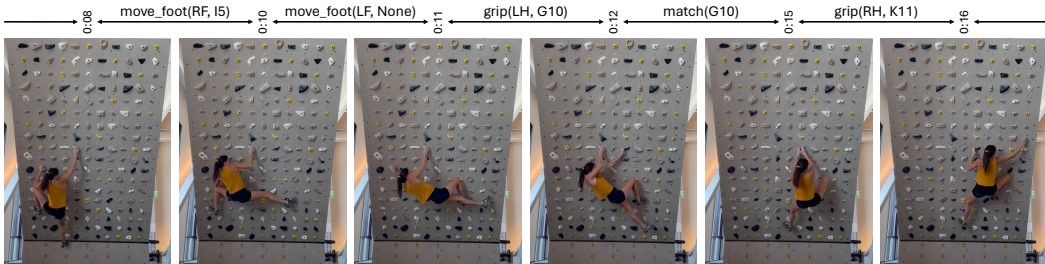

Figure 5: Example of video annotation: climbing videos are converted into structured action sequences, serving as ground-truth for evaluating LLM plans.

for models to predict. For instance, GPT-4o consistently achieves higher F1 and normalized LCS scores when evaluated on hand-only plans compared to full-body plans. However, this also highlights a critical limitation: real-world climbing heavily depends on footwork, which plays a central role in maintaining balance, reach, and efficient transitions.

While hand prediction serves as a useful lower bound on LLM capability, closing the gap between hand-only and full-body planning remains an open challenge. To fully model embodied reasoning in

Table 5: Personalized performance on EmbodiedPlan across different difficulty levels and agents.

| Level | Holds | Agent 1 | | Agent 2 | | Agent 3 | | Agent 2 → 1 | | Agent 3 → 1 | |
|-------|-------|---------|------|---------|------|---------|------|------|-------|------|-------|
| | | Actions | Norm. | Actions | Norm. | Actions | Norm. | LCS | Norm. | LCS | Norm. |
| V3 | 6.3 | 19.3 | 3.2 | 23.5 | 3.9 | 17.8 | 2.9 | 15.3 | 0.649 | 14.0 | 0.716 |
| V4 | 7.5 | 23.0 | 3.1 | 27.0 | 3.6 | 21.0 | 2.8 | 19.0 | 0.714 | 19.5 | 0.848 |
| V5 | 7.5 | 23.0 | 3.1 | 26.5 | 3.5 | 20.5 | 2.7 | 16.0 | 0.609 | 16.0 | 0.696 |
| V6 | 8.5 | 24.0 | 2.9 | 24.0 | 2.8 | 22.5 | 2.7 | 14.5 | 0.581 | 19.0 | 0.794 |
| Total | 7.2 | 21.7 | 3.1 | 24.9 | 3.5 | 19.9 | 2.8 | 16.0 | 0.641 | 16.5 | 0.754 |

Table 6: Personalized performance on EmbodiedPlan across different difficulty levels and agents (hands-only).

| Level | Holds | agent 1 | | agent 2 | | agent 3 | | agent 2 → 1 | | agent 3 → 1 | |
|-------|-------|---------|------|---------|------|---------|------|------|-------|------|-------|
| | | Actions | Norm. | Actions | Norm. | Actions | Norm. | LCS | Norm. | LCS | Norm. |
| V3 | 6.3 | 9.3 | 1.5 | 10.8 | 1.8 | 10.0 | 1.6 | 7.3 | 0.675 | 8.0 | 0.806 |
| V4 | 7.5 | 9.5 | 1.3 | 11.0 | 1.5 | 9.5 | 1.3 | 9.0 | 0.817 | 9.5 | 1.000 |
| V5 | 7.5 | 11.0 | 1.5 | 12.5 | 1.7 | 11.0 | 1.5 | 10.0 | 0.801 | 10.5 | 0.955 |
| V6 | 8.5 | 11.0 | 1.3 | 12.5 | 1.5 | 10.0 | 1.2 | 8.5 | 0.683 | 10.0 | 0.908 |
| Total | 7.2 | 10.0 | 1.4 | 11.5 | 1.6 | 10.1 | 1.4 | 8.4 | 0.730 | 9.2 | 0.895 |

climbing – and similar physically grounded tasks – future LLMs need to improve their understanding of lower-body coordination and its interaction with hand movements to achieve the goal.

## C  MORE CASE STUDIES

To better understand the strengths and limitations of LLM-generated plans, we present qualitative case studies comparing model outputs to ground-truth annotations. Figure 6 shows a side-by-side visualization of symbolic action sequences for one selected route, comparing plans generated for Agent 1, Agent 2, and Agent 3 against the human-annotated ground truth. Complementing this visualization, Table 9, 10, and 11 present the full data that is summarized in the main paper's case study (Table 3).

Table 7: Performance of various LLMs on EmbodiedPlan for standard agent (Agent 1) (hands-only).

| Level | Model | Validity (↑) | Actions | Precision | Recall | F1 (↑) | LCS | Norm. LCS (↑) |
|---|---|---|---|---|---|---|---|---|
| V3 | GPT-4o | 0.5 | 9.0 | 0.708 | 0.688 | 0.697 | 6.3 | 0.688 |
| | Gemini 2.5 Pro | 0.5 | 10.3 | 0.564 | 0.598 | 0.577 | 5.5 | 0.545 |
| | Claude 3.7 Sonnet | 0.3 | 9.0 | 0.565 | 0.548 | 0.556 | 5.0 | 0.548 |
| | Grok 3 | 0.8 | 9.0 | 0.708 | 0.685 | 0.696 | 6.3 | 0.685 |
| V4 | GPT-4o | 1.0 | 9.5 | 0.744 | 0.744 | 0.744 | 6.5 | 0.694 |
| | Gemini 2.5 Pro | 1.0 | 9.5 | 0.783 | 0.783 | 0.783 | 7.5 | 0.783 |
| | Claude 3.7 Sonnet | 1.0 | 9.5 | 0.644 | 0.644 | 0.644 | 6.0 | 0.644 |
| | Grok 3 | 1.0 | 9.5 | 0.672 | 0.672 | 0.672 | 6.5 | 0.672 |
| V5 | GPT-4o | 1.0 | 10.5 | 0.809 | 0.773 | 0.790 | 8.5 | 0.773 |
| | Gemini 2.5 Pro | 0.0 | 9.5 | 0.789 | 0.682 | 0.731 | 7.5 | 0.682 |
| | Claude 3.7 Sonnet | 0.0 | 10.5 | 0.568 | 0.545 | 0.556 | 6.0 | 0.545 |
| | Grok 3 | 1.0 | 10.5 | 0.714 | 0.682 | 0.697 | 7.5 | 0.682 |
| V6 | GPT-4o | 1.0 | 10.0 | 0.944 | 0.858 | 0.899 | 9.5 | 0.858 |
| | Gemini 2.5 Pro | 1.0 | 11.0 | 0.592 | 0.592 | 0.592 | 6.5 | 0.592 |
| | Claude 3.7 Sonnet | 0.5 | 11.0 | 0.367 | 0.367 | 0.367 | 4.0 | 0.367 |
| | Grok 3 | 0.5 | 11.0 | 0.752 | 0.733 | 0.741 | 8.0 | 0.708 |
| Total | GPT-4o | 0.8 | 9.6 | 0.783 | 0.750 | 0.766 | 7.4 | 0.740 |
| | Gemini 2.5 Pro | 0.6 | 10.1 | 0.658 | 0.651 | 0.652 | 6.5 | 0.629 |
| | Claude 3.7 Sonnet | 0.4 | 9.8 | 0.542 | 0.530 | 0.536 | 5.2 | 0.530 |
| | Grok 3 | 0.8 | 9.8 | 0.711 | 0.692 | 0.700 | 6.9 | 0.687 |

Table 8: Personalized planning performance of LLMs on EmbodiedPlan for Agent 2 and 3 (hands-only).

| agent | Model | Validity (↑) | Actions | Precision | Recall | F1 (↑) | LCS | Norm. LCS (↑) | LCS → agent 1 | Norm. LCS → agent 1 |
|---|---|---|---|---|---|---|---|---|---|---|
| agent 2 | GPT-4o | 0.8 | 9.6 | 0.746 | 0.624 | 0.679 | 7.0 | 0.616 | 9.1 | 0.946 |
| | Gemini 2.5 Pro | 0.6 | 10.3 | 0.677 | 0.614 | 0.640 | 6.9 | 0.597 | 6.4 | 0.610 |
| | Claude 3.7 Sonnet | 0.4 | 10.1 | 0.518 | 0.459 | 0.486 | 5.2 | 0.459 | 8.6 | 0.852 |
| | Grok 3 | 0.8 | 9.8 | 0.642 | 0.545 | 0.589 | 6.3 | 0.545 | 8.5 | 0.869 |
| agent 3 | GPT-4o | 0.7 | 9.8 | 0.769 | 0.744 | 0.754 | 7.4 | 0.720 | 8.7 | 0.886 |
| | Gemini 2.5 Pro | 0.7 | 9.7 | 0.764 | 0.744 | 0.749 | 7.3 | 0.699 | 6.9 | 0.663 |
| | Claude 3.7 Sonnet | 0.4 | 9.8 | 0.502 | 0.486 | 0.493 | 4.9 | 0.479 | 8.1 | 0.815 |
| | Grok 3 | 0.6 | 10.1 | 0.615 | 0.605 | 0.607 | 6.2 | 0.580 | 7.7 | 0.764 |

Table 9: A case study of model performance on a single challenging route (Agent 1).

| Model | Actions (Human: 26) | Precision | Recall | F1 (↑) | LCS | Norm. LCS (↑) | CoG Length (Human: 22.2) |
|---|---|---|---|---|---|---|---|
| GPT-4o | 25 | 0.680 | 0.654 | 0.667 | 13 | 0.500 | 22.1 |
| Gemini 2.5 Pro | 29 | 0.448 | 0.500 | 0.473 | 10 | 0.345 | 22.8 |
| Claude 3.7 Sonnet | 24 | 0.458 | 0.423 | 0.440 | 7 | 0.269 | 26.5 |
| Grok 3 | 27 | 0.593 | 0.615 | 0.604 | 7 | 0.259 | 24.8 |

Table 10: A case study of model performance on a single challenging route (Agent 2).

| Model | Actions (Human: 28) | Precision | Recall | F1 (↑) | LCS | Norm. LCS (↑) | | | CoG Length (Human: 23.3) |
|---|---|---|---|---|---|---|---|---|---|
| GPT-4o | 25 | 0.640 | 0.571 | 0.604 | 13 | 0.464 | 16 | 0.640 | 22.1 |
| Gemini 2.5 Pro | 27 | 0.481 | 0.464 | 0.473 | 9 | 0.321 | 15 | 0.517 | 24.6 |
| Claude 3.7 Sonnet | 24 | 0.542 | 0.464 | 0.500 | 11 | 0.393 | 13 | 0.542 | 26.5 |
| Grok 3 | 27 | 0.667 | 0.643 | 0.655 | 16 | 0.571 | 19 | 0.704 | 24.8 |

Table 11: A case study of model performance on a single challenging route (Agent 3).

| Model | Actions (Human: 26) | Precision | Recall | F1 (↑) | LCS | Norm. LCS (↑) | | | CoG Length (Human: 22.1) |
|---|---|---|---|---|---|---|---|---|---|
| GPT-4o | 26 | 0.615 | 0.615 | 0.615 | 13 | 0.500 | 16 | 0.615 | 22.4 |
| Gemini 2.5 Pro | 28 | 0.607 | 0.654 | 0.630 | 13 | 0.464 | 17 | 0.586 | 22.4 |
| Claude 3.7 Sonnet | 26 | 0.423 | 0.423 | 0.423 | 11 | 0.423 | 17 | 0.654 | 26.5 |
| Grok 3 | 24 | 0.625 | 0.577 | 0.600 | 14 | 0.538 | 18 | 0.667 | 24.8 |

| Ground Truth Plan | Agent 1 Plan | Agent 2 Plan | Agent 3 Plan |
|---|---|---|---|
| grip(LH, A4) | grip(LH, A4) | grip(LH, A4) | grip(LH, A4) |
| grip(RH, D3) | grip(RH, D3) | grip(RH, D3) | grip(RH, D3) |
| move_foot(LF, chip) | move_foot(LF, chip) | move_foot(LF, chip) | move_foot(LF, chip) |
| move_foot(RF, None) | move_foot(RF, chip) | move_foot(RF, chip) | move_foot(RF, chip) |
| grip(LH, B8) | grip(LH, B8) | move_foot(LF, A4) | grip(LH, I5) |
| move_foot(RF, chip) | move_foot(LF, A4) | grip(RH, I5) | move_foot(RF, D3) |
| move_foot(LF, None) | move_foot(RF, D3) | move_foot(RF, D3) | move_foot(LF, A4) |
| grip(RH, E10) | grip(RH, E10) | grip(LH, B8) | grip(RH, B8) |
| move_foot(LF, A4) | move_foot(LF, B8) | move_foot(LF, B8) | move_foot(LF, I5) |
| move_foot(RF, I5) | move_foot(LF, chip) | move_foot(RF, A4) | grip(LH, E10) |
| move_foot(LF, None) | grip(LH, G10) | grip(RH, G10) | move_foot(RF, B8) |
| grip(LH, G10) | move_foot(LF, E10) | move_foot(RF, I5) | grip(RH, G10) |
| match(G10) | move_foot(RF, chip) | grip(LH, E10) | move_foot(LF, E10) |
| grip(RH, K11) | grip(RH, K11) | move_foot(LF, E10) | grip(LH, K11) |
| move_foot(LF, I5) | move_foot(RF, G10) | move_foot(RF, G10) | move_foot(RF, G10) |
| move_foot(RF, None) | move_foot(LF, chip) | grip(RH, K11) | dynamic(RH, I14) |
| grip(LH, I14) | grip(LH, I14) | move_foot(LF, B8) | move_foot(LF, K11) |
| move_foot(RF, I5) | move_foot(LF, E10) | move_foot(RF, G10) | move_foot(RF, None) |
| move_foot(LF, None) | move_foot(RF, K11) | grip(LH, I14) | grip(LH, I16) |
| grip(RH, I16) | grip(RH, I16) | move_foot(LF, E10) | move_foot(RF, I14) |
| move_foot(LF, E10) | move_foot(RF, I14) | move_foot(RF, I5) | move_foot(LF, None) |
| move_foot(RF, K11) | move_foot(LF, G10) | grip(LH, I16) | dynamic(RH, F18) |
| move_foot(LF, G10) | grip(LH, F18) | move_foot(LF, I14) | match(F18) |
| grip(LH, F18) | match(F18) | move_foot(RF, G10) | top_out() |
| match(F18) | top_out() | dynamic(RH, F18) | |
| top_out() | | move_foot(LF, I16) | |
| | | move_foot(RF, I14) | |
| | | match(F18) | |
| | | top_out() | |

Figure 6: Side-by-side comparison of the Ground Truth Plan and the generated plans of Agent 1, Agent 2, and Agent 3.