# OpenReview forum: "Probing the Limits of Embodied Spatial Planning in LLMs"
_ICLR.cc/2026/Conference — ICLR 2026 Conference Withdrawn Submission_

### Official Review · Reviewer_SaLW · 2025-10-21

**Soundness:** 2
**Presentation:** 3
**Contribution:** 2
**Rating:** 2
**Confidence:** 4

**Summary:**

This paper introduces EmbodiedPlan, a benchmark designed to evaluate the embodied spatial planning capabilities of Large Language Models (LLMs) in physically grounded tasks. Using indoor bouldering as a testbed, the authors assess LLMs’ ability to generate symbolic action plans that respect physical constraints and agent-specific embodiment. The benchmark includes multiple agent profiles and a suite of evaluation metrics, including syntactic validity, semantic alignment, and center-of-gravity (CoG) trajectory simulation. The empirical results reveal a significant gap between syntactic fluency and physical plausibility in current LLMs, highlighting limitations in spatial imagination and embodied reasoning.

**Strengths:**

1. **Well-Motivated Problem**: The paper is well-motivated and addresses a critical gap in current LLM capabilities—namely, their lack of grounded physical reasoning. This is highly relevant for deploying AI agents in real-world environments.

2. **Comprehensive Evaluation**: The benchmark includes a variety of metrics and agent profiles (e.g., height of the agents), revealing clear limitations in current LLMs' ability to adapt plans to physical constraints.

3. **Clear Writing and Presentation**: The paper is well-organized and clearly written, with informative figures and tables that support the claims.

**Weaknesses:**

1. **Oversimplified Physical Modeling**:

    - The benchmark treats the human body as a point mass, ignoring rotational dynamics and gravity distribution. This simplification undermines the realism of the task and diverges from the paper’s stated motivation (physical-aware reasoning).
    - For example, the rotational angle of the body significantly affects force distribution between limbs. Hands typically bear less weight than legs, and this biomechanical reality is not modeled.
    - Consequently, the task reduces to a symbolic path-finding problem rather than a physically grounded planning challenge.

2. **Evaluation Criteria Ambiguity**:

    - Precision and Recall: These metrics are typically suited for offline datasets. It’s unclear how they are computed in an online decision-making context.
    - Center-of-Gravity (CoG): CoG is defined using only hand positions. This neglects the role of foot placement and body orientation, which are critical in climbing and embodied movement.
    - Lack of Online Evaluation Metric: Existing metrics (e.g., LCS) measure similarity to ground truth but fail to capture online performance or feasibility. There is no mechanism to evaluate whether a generated plan is executable or stable in real-time. This problem should be considered since there could have multiple solutions for a single task.

3. **Divergence from Motivation**:

    - The benchmark’s simplifications result in a mismatch between the paper’s motivation (testing embodied reasoning under physical contraints). This weakens the contribution and raises concerns about the benchmark’s validity.

**Questions:**

**Questions**:  See Weakness

**Suggestions**:
The motivation behind this work is strong and timely. However, the current benchmark design oversimplifies the physical modeling, which limits its effectiveness. If this problem is addressed, this work has the potential to become a valuable and impactful benchmark for embodied reasoning in LLMs.

---

### Official Review · Reviewer_Bssf · 2025-10-30

**Soundness:** 3
**Presentation:** 3
**Contribution:** 3
**Rating:** 4
**Confidence:** 3

**Summary:**

EmbodiedPlan is a new benchmark that tests LLMs on indoor-climbing plans. Given route coordinates and agents' heights, an LLM outputs a sequence of discrete actions (grip, match, move foot). Performance is scored on (1) symbolic correctness, (2) semantic alignment to expert plans, and (3) physical plausibility via a simplified center-of-gravity simulator. GPT-4o, Gemini Pro, etc. write valid-looking moves but often ignore real-world physics, especially for short climbers—showing current LLMs still lack true physical reasoning.

**Strengths:**

1.  An interesting real world task: indoor bouldering as a discrete symbolic planning task, extending evaluation of LLM beyond text-based games.
2.  Evaluation from multiple views (symbolic, semantic, physical) that captures nuances of embodied reasoning.
3.  Take personalization and embodiment as a condition, which is a rarely explored dimension.

**Weaknesses:**

1.  Only 2-D MoonBoard walls are considered; no evidence the findings generalize to richer 3-D or non-climbing tasks.
2.  Datasets are not very easy to get since it is constructed from real videos.
3. Several SOTA LLMs such as  GPT-5 are missing in some experiments.

**Questions:**

1.  What is the exact size and diversity (number of routes, difficulty grades, agent profiles) of EmbodiedPlan?
2.  Can this benchmark extend to 3-D to capture more realistic scenarios?

---

### Official Review · Reviewer_EHWH · 2025-10-31

**Soundness:** 2
**Presentation:** 2
**Contribution:** 2
**Rating:** 2
**Confidence:** 3

**Summary:**

This paper proposes a benchmark to evaluate LLMs' ability to generate an actionable bouldering plan that respect geometric, physical and bodily constraints. Three variations of agent profiles are included to test personalized planning. However, there are serious flaws in the benchmark design and evaluation method.

**Strengths:**

1. The paper is clearly written and easy to follow.
2. The choice of the bouldering planning task to benchmark LLMs' planning ability is indeed intriguing.

**Weaknesses:**

1. While bouldering is a good sport that gains popularity in modern life, I am not convinced it is a good choice to test the planning abilities of LLMs. As an amateur who tried indoor bouldering a few times, the challenge of bouldering does not lie in route planning. Due to the large variety of possible routes and difference in climber's skill levels, there is no such a thing as "optimal plan" for bouldering. That is why in bouldering competition the points are gained as long as the top is reached, regardless of the plan taken. Therefore, I found it questionable to use Gemini-generated plans as ground-truth or any "ground-truth" plans to evaluate the embodied planning ability of other LLMs for this specific task. If only validity of plans is considered, the benchmark becomes trivial as even small open-sourced model such as Qwen3-4B can produce flawless plans.
2. The metrics for LLM evaluations rely heavily on comparison with the ground-truth plans labelled by Gemini and reviewed by humans, which do not really reflect the spatial imagination and reasoning ability of LLMs. For example, in Table 1, Qwen3-4B has better LCS compared to GPT-4o-mini, which is far from my personal experience with these two models.

**Questions:**

1. What are the results of the most powerful LLMs like GPT-4o, Gemini, Claude and Grok in the main benchmark as shown in Table 1?

---

### Official Review · Reviewer_BGpb · 2025-11-06

**Soundness:** 3
**Presentation:** 2
**Contribution:** 2
**Rating:** 6
**Confidence:** 3

**Summary:**

This paper explores an interesting question: can the symbolic reasoning capabilities of Large Language Models (LLMs) be extended to the physical world? By introducing a novel indoor bouldering task and constructing a dedicated EmbodiedPlan dataset, this paper conducts multi-dimensional experimental evaluations. The results demonstrate that current SOTA LLMs lack grounded physical imagination.

**Strengths:**

1.	The problem explored in this paper is interesting, and the task setup is easy to follow.
2.	This paper defines agents with different heights and introduces gravity as a physical constraint to simulate a real-world physical environment.
3.	This paper evaluates the performance of LLMs on this task across five different dimensions, providing a relatively comprehensive analysis.

**Weaknesses:**

1.	The physical environment in this paper is highly simplified. For example, the Center-of-Gravity of the agent is approximated by averaging the coordinates of its two hands. Moreover, in real-world scenarios, considering only a 2D physical environment is clearly insufficient.
2.	The definition of agent attributes is relatively limited, as it only considers height and gender, without taking into account other fine-grained characteristics. For example, arm span is a crucial physical attribute in bouldering, and different arm spans may lead to entirely different route-planning strategies. Therefore, future work could consider evaluating the spatial planning capabilities of LLMs using multi-dimensional physical characteristics
3.	Although this paper explores the problem in depth, it does not further investigate the underlying causes, nor does it discuss whether existing techniques can effectively address this limitation. Whether this issue can be alleviated by incorporating an external physics engine or by constructing similar embodied sequence data.

**Questions:**

Please see the weaknesses.

---

### Note · Authors · 2026-01-08

I have read and agree with the venue's withdrawal policy on behalf of myself and my co-authors.